# Co-Application of 1-MCP and Laser Microporous Plastic Bag Packaging Maintains Postharvest Quality and Extends the Shelf-Life of Honey Peach Fruit

**DOI:** 10.3390/foods11121733

**Published:** 2022-06-14

**Authors:** Xuerui Li, Sijia Peng, Renying Yu, Puwang Li, Chuang Zhou, Yunhui Qu, Hong Li, Haibo Luo, Lijuan Yu

**Affiliations:** 1Agro-Products Processing Research Institute, Yunnan Academy of Agricultural Sciences, Kunming 650221, China; lxr@yaas.org.cn (X.L.); quyunhui404@163.com (Y.Q.); ynveg@163.com (H.L.); 2School of Food Science and Pharmaceutical Engineering, Nanjing Normal University, Nanjing 210023, China; psjxuan@163.com (S.P.); 18606621900@163.com (R.Y.); 3South Subtropical Crop Research Institute of Chinese Academy of Tropical Agricultural Sciences, Key Laboratory of Hainan Province for Postharvest Physiology and Technology of Tropical Horticultural Products, Zhanjiang 524091, China; puwangli@163.com (P.L.); zhouchuang0759@163.com (C.Z.)

**Keywords:** honey peach fruit, soft, rot, preservation, shelf-life

## Abstract

Honey peach (*Prunus persica* L.) is highly nutritious; it is an excellent source of sugars, proteins, amino acids, vitamins, and mineral elements. However, it is a perishable climacteric fruit that is difficult to preserve. In this study, “Feicheng” honey peach fruit was used as a test material to investigate the synergistic preservation effect of 1-methylcyclopropene (1-MCP) and laser microporous film (LMF). The peach fruits were fumigated for 24 h with 2 μL L^−1^ 1-MCP, then packed in LMF. In comparison with the control treatment, 1-MCP + LMF treatment markedly decreased the respiration rate, weight loss, and rot rate of peach fruits. Moreover, the combination of 1-MCP and LMF suppressed the increase in soluble solids (SS) and reducing sugars (RS), as well as the decrease in titratable acid (TA) and ascorbic acid (AsA). The combined application also maintained a high protopectin content and low soluble pectin content; it reduced the accumulation of superoxide anions (O_2_^−^) and hydrogen peroxide (H_2_O_2_). Except in a few samples, the catalase (CAT) and ascorbate peroxidase (APX) activities were higher when treated by 1-MCP + LMF. Conversely, the phenylalanine deaminase (PAL), peroxidase (POD), lipase, lipoxygenase (LOX), polygalacturonase (PG), β-glucosidase, and cellulase (Cx) activities were lower than in the control. Furthermore, 1-MCP + LMF treatment reduced the relative abundances of dominant pathogenic fungi (e.g., *Streptomyces*, *Stachybotrys*, and *Issa* sp.). The combined treatment improved the relative abundances of antagonistic fungi (e.g., *Aureobasidium* and *Holtermanniella*). The results indicated that the co-application of 1-MCP and LMF markedly reduced weight loss and spoilage, delayed the decline of nutritional quality, and inhibited the physiological and biochemical metabolic activities of peach during storage. These changes extended its shelf-life to 28 days at 5 °C. The results provide a reference for the commercial application of this technology.

## 1. Introduction

*Prunus persica* L., which pertains to the Rosaceae family, is cultured worldwide. Its fruit (i.e., peach) is an important horticultural product [1]. Because of its pleasant aroma, juicy texture, delicate flavor, and rich nutrient content, peach is one of the favorite fruits worldwide [2]. Honey peach is an important soft melting type of peach fruit. However, fresh honey peach fruits are extremely perishable during storage because of their vigorous respiratory physiological metabolism, rapid softening, and high moisture content; these characteristics are favorable for the growth of pathogenic microorganisms [3,4]. Thus, there is a need to explore suitable preservation techniques to reduce honey peach fruit rot, extend its shelf-life, and improve its economic benefits.

Refrigeration, heat treatment, irradiation, air conditioning, methyl jasmonate or 1-MCP, edible coatings, and biological preservation have been used to store honey peaches [4,5]. Among these preservation technologies, 1-MCP fumigation and refrigeration are frequently practiced because of their superior features, such as ease of operation and efficiency [6]. Wang et al. [7] found that 1-MCP delayed the ripening of peach fruit at room temperature. Liu et al. [8] used 1-MCP to fumigate peach fruit at 20 °C for 10 days; they found that 5 μL L^−1^ 1-MCP effectively delayed ethylene production and the respiration rate of peach fruit. 1-MCP blocks the ripening by binding to ethylene receptors. However, the de novo synthesis of ethylene receptors could occur with the progress of storage time. Other technologies are needed to improve its fresh-keeping effect in the later period of storage, such as refrigeration, film coating, and packaging [5].

Recently, microporous film packaging has emerged as an environmentally sustainable technology. Micropores on the film improve its permeability; they effectively adjust humidity and the ratio of O_2_ and CO_2_ in the packaging [9]. Liguori et al. [10] used microporous film to pack grapes; they found that it significantly inhibited the decay of the fruit and prolonged its storage at 5 °C. Chiabrando et al. [11] showed that the microporous film packaging maintained the color, acidity, and vitamin C content of fresh strawberries and grapes while reducing weight loss and rot during storage at 6 °C. Cliff et al. [12] found that fresh-cut apples packed in microporous film had a better hardness and sensory quality at the end of storage. In addition, there was evidence that the combination of microporous film packaging with other preservation technologies (e.g., ozone, UV-C, citric acid, ascorbic acid, and an ethylene adsorbent) has a synergistic effect [9]. Rodriguez and Zoffoli [13] treated blueberry fruit with SO_2_ and packed them in perforated polyethylene bags, which effectively inhibited fruit decay and prolonged the quality. Villalobos et al. [14] combined defatted soybean meal extract with microporous film packaging, which reduced mold and yeast levels and extended the storage life of figs.

A laser microporous film (LMF) was punched with a specific number of micron scale holes according to the respiration rates of different fruits and vegetables using a laser perforating system. The LMF spontaneously adjusted the dynamic balance of gas and water vapor in the package; it provided better air permeability than traditional packaging did [9]. Based on a sensory evaluation, our preliminary experiments showed that the LMF retarded fruit rot and maintained the appearance of honey peach fruits under ambient temperature storage. However, the synergistic effects of 1-MCP and LMF on preserving fresh honey peach fruits remain poorly understood.

In this study, Shandong Feicheng honey peach was used as the test material to evaluate the synergistic effect of 1-MCP + LMF treatment on the preservation of fresh honey peach fruits. The study specifically addressed the effects of 1-MCP + LMF on changes in quality, physiology, biochemical metabolism, and microbial community structure. The results will provide useful technical support for maintaining postharvest quality and extend the shelf-life of honey peach fruit.

## 2. Materials and Methods

### 2.1. Plant Materials

Honey peach fruits at 80% maturity were collected from the Feicheng Orchard in Tai’an City, Shandong Province in early September 2020 (16–27 °C). They were pre-cooled at 5 °C to dissipate the field heat. Healthy peach fruits, uniform maturity, free of pests, disease, and mechanical damage, were selected as the experimental materials.

### 2.2. Experimental Treatments

The selected peach fruits were randomly divided into two groups (105 peaches per group). The first group was fumigated in a polystyrene sealed box with 2 μL L^–1^ 1-MCP (Xianyang Xiqin Biological Technology Co., Ltd., Xi’an, China) for 24 h at 5 °C. After fumigation, five peaches were packed in one LMF bag (length × width = 45 × 30 cm; Beijing Dupack Packaging Technology Co., Ltd., Beijing, China), made of polyethylene with a thickness of 80 μm, pore size of 100 μm, and 20 holes. The second group was used as a control; peaches in this group were sealed for 24 h at 5 °C without fumigation, then packaged into double corrugated boxes. Subsequently, the samples were stored at 5 °C, and 15 peach fruits (about 3 kg) were removed at 7-day intervals for a related index determination. The optimal 1-MCP concentration and packaging method were selected in view of preliminary research (Appendix A in Appendix A).

### 2.3. Determination of Weight Loss, Rot Rate, and Respiration Rate

The weight loss was calculated using the formula: WL (%) = 100 × (W_0_ − Wt)/W_0_. The initial weight of the sample after harvest was recorded as W_0_; the weight of each sampling point was recorded as Wt. The rot rate was calculated using the formula: rot rate (%) = number of rotten fruits × 100/total number of stored fruits. The total number of stored fruits was 105 for each treatment. Respiratory rate was measured using a SYS-GH 30A respiration meter (Liaoning Saiyasi Technology Co., Ltd., Shenyang, China). The result was expressed as the milligrams of carbon dioxide produced per kilogram in an hour (mg CO_2_ kg^−1^ h^−1^).

### 2.4. Determination of the Soluble Solid (SS), Reducing Sugar (RS), Titratable Acid (TA), Ascorbic Acid (AsA), Protopectin, and Soluble Pectin Contents

The SS content was measured using an AWAJ refractometer (Shanghai Yehuo Instrument Co., Ltd., Shanghai, China). The RS content was determined with method described by La et al. [15] and Gouda et al. [16]. Briefly, 2 g of sample were blended with 5 mL of ddH_2_O, and then centrifuged for 20 min at 10,000× *g*. Next, 0.1 mL supernatant was mixed with 2 mL of 3,5-dinitrosalicylic acid and incubated for 5 min at 100 °C. After cooling, the absorption at 540 nm (A_540_) was estimated.

The TA content was determined as the description by Gouda et al. [16]. 10 g of sample was crushed and homogenized with 90 mL of boiled distilled water (pH 8.3). The mixture was centrifuged for 10 min (10,000× *g*, 4 °C). 20 mL of the supernatant was titrated to pH 8.3 with 0.01 N NaOH. The TA content was calculated as grams of malic acid per kilogram according to the fresh weight. The AsA content was measured with the method described by Zhao et al. [17], and the AsA content was calculated as grams of AsA per kilogram according to the fresh weight.

The protopectin and soluble pectin contents were measured as per the description by Tang et al. [18]. Briefly, 1 g of peach powder was added to 25 mL of 95% ethanol, heated with boiling water for 30 min, and then centrifuged for 15 min (8000× *g*, 4 °C). The precipitate was added to 25 mL of 90% ethanol, then incubated in boiling water for 30 min and repeated three times. The mixture was quickly cooled, then centrifuged at 8000 × *g* for 15 min. Subsequently, the precipitate was mixed with 30 mL of deionized water, and was then shaken at 50 °C for 35 min; next, it was cooled and centrifuged for 15 min (8000× *g*, 4 °C). The supernatant was collected to determine soluble pectin content. For protopectin extraction, the precipitate was added to 25 mL of 0.5 M sulfuric acid solution and shaken with boiling water for 1.5 h; it was then cooled and centrifuged for 15 min (8000× *g*, 4 °C). For further analysis, the supernatant was collected. The reaction mixture of 1 mL of protopectin or soluble pectin extracts with 6 mL of concentrated sulfuric acid was heated in a boiling water bath for 30 min; next, it was cooled and added to 0.2 mL of 1.5 g L^−1^ carbazole-ethanol. After incubating for 30 min in the dark, the absorption A_5__30_ was estimated to calculate the content, which was expressed as grams of galacturonic acid per kilogram based on the fresh weight.

### 2.5. Determination of the Superoxide Anion (O_2_^−^) and Hydrogen Peroxide (H_2_O_2_) Contents

The O_2_^−^ production rate and H_2_O_2_ content were analyzed as the report by Gouda et al. [16], with part adjustment. First, 2 g of ground peach was mixed with 5 mL of 65 mM potassium phosphate buffer solution (PBS) (pH 7.8), 1 mL of 10 mM hydroxylamine hydrochloride, and 1 mL of 0.1 M ethylenediamine tetraacetic acid; the mixture was centrifuged at 4 °C for 15 min (10,000× *g*). After centrifugation, 2 mL of the supernatant was mixed with 2 mL of 7 mM α-naphthylamine and 2 mL of 17 mM p-aminobenzenesulfonic acid then mixed with 3 mL of anhydrous ether after incubating at 37 °C for 15 min. After centrifugation, the water phase was removed and the A_530_ value was measured to calculate the O_2_^−^ production rate based on the fresh weight.

To determine the H_2_O_2_ content, 2 g of ground peach was homogenized with 6 mL of pre-cooled acetone then centrifuged at 4 °C for 15 min with the speed of 10,000× *g*. Subsequently, 4 mL of the supernatant was added with 0.2 mL of 5% titanium sulfate and 0.4 mL of concentrated ammonia to centrifuge again for 10 min (3000× *g*). The precipitate was washed three times with acetone, then dissolved in 5 mL of 2 M sulfuric acid. After dissolution, the volume was adjusted to 10 mL to measure the A_415_ value. The H_2_O_2_ content was expressed as millimolar H_2_O_2_ per kilogram according to the fresh weight.

### 2.6. Assays of Enzyme Activity

#### 2.6.1. Phenylalanine Ammonia Lyase (PAL), Peroxidase (POD), Catalase (CAT), and Ascorbate Peroxidase (APX) Activities

Each enzyme was extracted under 4 °C. For PAL, 2 g of the peach fruit powder was added to 6 mL of 100 mM sodium borate buffer (pH 8.7, containing 20 mM of β-mercaptoethanol and 1% polyvinyl-pyrrolidone). For POD, the peach fruit powder was mixed with 6 mL of 50 mM sodium borate buffer (pH 8.7). For CAT and APX, 2 g of peach fruit powder were blended with 6 mL of 50 mM PBS (pH 7.8, pre-cooled, containing 0.1 mM ethylenediamine tetraacetic acid, 1 mM ascorbic acid, and 1% polyvinyl-pyrrolidone). After that, the extracts were homogenized and centrifuged for 15 min (10,000× *g*, 4 °C). The supernatants were collected for enzyme assays.

To assay the PAL, POD, CAT, and APX activity, the method described by Gouda et al. [16] were conducted. The amount of enzyme that caused an increase in absorbance of 0.01 at 290 nm in 1 h was used to express PAL activity, and the amount of enzyme that caused an increase in absorbance of 0.001 at 460 nm in 1 min was used to express POD activity; for CAT and APX, single units of enzyme activity were defined as U g^−1^ with a 0.01 change in the A_240_ and A_290_ values per minute, respectively.

#### 2.6.2. Lipase and Lipoxygenase (LOX) Activities

Lipase activity was measured as the study by Liu et al. [19], with some modifications. Briefly, 5 g of ground peach was added with 10 mL of 200 mM PBS (pH 7.8, pre-cooled, containing 50 mM β-mercaptoethanol) then centrifuged for 20 min (15,000× *g*, 4 °C). After that, 0.5 mL of the supernatant was mixed with 2.3 mL of 200 mM PBS (pH 7.8) and 0.5 mL of 0.5% α-naphthyl acetate solution, then incubated for 40 min (30 °C) and added with 0.2 mL of 0.15% fast blue B salt containing 6% sodium lauryl sulfate. After 5 min, the A_520_ value was measured, and lipase activity was calculated based on the change of 0.01 in the A_520_ value per minute.

LOX activity was determined as the description by Liu et al. [19], with part modifications. Briefly, 2 g of the sample powder was added to 6.0 mL of 50 mM PBS (pre-cooled) and then centrifuged at 4 °C for 15 min with the speed of 10,000× *g*. Then, 0.1 mL of the supernatant was added with 2.875 mL of the pH 7.6 phosphate-citric acid buffer and 0.1 mL of 0.25% linoleic acid. Changes in A_243_ values were recorded at 30-s intervals for 3–5 min. LOX activity were expressed as U g^−1^FW according on the change of 0.01 in A_243_ value per minute.

#### 2.6.3. Polygalacturonase (PG), β-Glucosidase, and Cellulase (Cx) Activities

The PG, β-glucosidase, and Cx activities were determined as the description by Tang et al. [18], with appropriate modifications.

The PG, β-glucosidase, and Cx activities were analyzed by mixing 10 g of ground peach with 20 mL of pre-cooled 90% ethanol. The mixture was then centrifuged for 20 min (12,000× *g*, 4 °C). The precipitate was added with 10 mL of 75% ethanol and 5 mL of 50 mM sodium acetate buffer (pH 5.5, pre-cooled, containing 1.8 M sodium chloride) was added. After incubation for 20 min at 4 °C, the mixture was centrifuged for 20 min (12,000× *g*, 4 °C). The reaction mixture composed of 1 mL of 50 mM sodium acetate buffer, 0.5 mL of the supernatant, 0.5 mL of 1% polygalacturonic acid solution for PG, 1.5 mL of 1% salicin solution for β-glucosidase, or 1.5 mL of 1% carboxymethyl cellulose solution for Cx; the mixture was incubated at 37 °C for 1 h. One unit of PG, β-glucosidase, and Cx was defined as the amount of enzyme that catalyzed the decomposition of 1% polygalacturonic acid, 1% salicin, or 1% carboxymethyl cellulose, respectively, with the formation of 1 μg of reaction product per hour.

### 2.7. Analyses of Microbial Community Structure

Microbial community structure analyses were performed by Sangon Biotech Co., Ltd. (Shanghai, China). Fungal and bacterial genomic DNA extraction were conducted with E.Z.N.A™ Mag-Bind Soil DNA Kit (Omega Bio-Tek Inc., Norcross, GA, USA) after being assessed with 1% agarose gel electrophoresis. These DNA samples were then used as templates to amplify the internal transcribed spacer (ITS) sequences for fungi, or the 16SrDNA sequences for bacteria. Samples were distinguished by attaching barcode sequences with different bases to the forward primer. The corresponding primers were as follows: ITS1 F: 5′-CTTGGTCATTTAGAGGAAGTAA-3′, ITS2 R: 5′-GCTGCGTTCTTCATCGATGC-3′; 341 F: 5′-CCTACGGGNGGCWGCAG-3′, and 805 R: 5′-GACTACHVGGGTATCTAATCC-3′.

After two rounds of polymerase chain reaction amplification, an Illumina MiSeq system (Illumina, San Diego, CA, USA) was used to sequence the amplification products. Cladistic analyses of fungal and bacterial genera were performed using the Blast and the Unite database (http://unite.ut.ee/index.php, accessed on 15 October 2020), or the Ribosomal Database Project classifier 2.12 (https://sourceforge.net/projects/rdp-classifier/, accessed on 15 October 2020) and the Ribosomal Database Project database (http://rdp.cme.msu.edu/misc/resources.jsp, accessed on 15 October 2020), respectively.

### 2.8. Statistical Analysis

SPSS 19.0 software was used to perform one-way analysis to identify significant differences among treatments. A least significant difference test was used to determine significant differences of the means at *p* < 0.05. Excel 2016 software was applied to draw figures.

## 3. Results

### 3.1. Changes in Appearance Quality

As shown in Figure 1, the fruits on day 0 were fresh with a small amount of red peel around the pedicle and green peach tip. Moreover, the fresh samples had a typical peach aroma. However, after storage at 5 °C for 35 days, peach fruits in the control group significantly softened, yellowed, shrank, rotted, and lost water; they also had a peculiar smell. These fruits had almost entirely lost commercial value. In contrast to the control group, the co-application of 1-MCP and LMF resulted in significantly better appearance and hardness; although the peel had slightly yellowed, the overall appearance had not significantly changed and the aroma had lightened. Based on the above findings at 5 °C, honey peach fruits generally could be stored for no more than 14 days, but the co-application treatment might prolong their shelf-life to 28 days.

### 3.2. Changes in Weight Loss, Rot Rate, and SS, RS, TA, AsA, Protopectin, and Soluble Pectin Contents

The weight loss, rot rate, and SS content in the control group increased sharply during storage, from 0%, 0%, and 9.33% on day 0 of storage to 26.53%, 16.67%, and 14.67%, respectively, on day 35 of storage (Figure 2A–C). Conversely, the co-application of 1-MCP and LMF significantly inhibited the respective enhancements of these parameters, with slight increases of 5.79%, 6.25%, and 10.49% after storing for 35 days (Figure 2A–C). The RS content in the control group first increased and then decreased, with a maximum of 213.18 g kg^−1^ at 28 days (Figure 2D). The RS content was markedly lower in the treatment group than in the control group at 28 days, but it was significantly higher at 35 days in the treatment group (Figure 2D). The TA and AsA contents in the control group constantly decreased during storage (Figure 2E,F). However, the co-application of 1-MCP and LMF apparently retarded the decreases in TA and AsA contents (Figure 2E,F). The protopectin content in the control group also fluctuated with an increase, decrease, and then a further increase. Conversely, the soluble pectin content rose rapidly until storing for 21 days, and after that it decreased. The co-application of 1-MCP and LMF maintained a high protopectin content and low soluble pectin content in peaches during storage (Figure 2G,H).

### 3.3. Changes in Respiration Rate, O_2_^−^ Production Rate, and H_2_O_2_ Content

The respiration rate, O_2_^−^ production rate, and H_2_O_2_ content under control conditions and with the co-application of 1-MCP and LMF were investigated (Figure 3). Although the respiration rate of the control group fluctuated irregularly, it reached a maximum value of 1193.06 μmol min^−1^ kg^−1^ after 7 days of storage, then remained high. However, the co-application of 1-MCP and LMF significantly inhibited the respiration of peach fruits and maintained it at a low level throughout the storage period (Figure 3A). The O_2_^−^ production rate and H_2_O_2_ content were generally consistent with the respiratory rate. The 1-MCP + LMF treatment significantly reduced the O_2_^−^ production rate and H_2_O_2_ content (Figure 3B,C).

### 3.4. Changes in PAL, POD, CAT, APX, Lipase, and LOX Activity

The PAL, POD, CAT, APX, lipase, and LOX activities of the control group peach fruits displayed an overall upward trend during storage (Figure 4). The LOX activity increased from 238.7 U g^−1^ at 0 days to 2706.7 U g^−1^ after 21 days, and 66,866.7 U g^−1^ after 35 days, respectively: these constituted increases of 11.34-fold and 280.13-fold (Figure 4F). Compared with the control treatment, 1-MCP + LMF treatment maintained the activities of PAL, POD, lipase, and LOX; it increased the activities of CAT and APX (Figure 4C,D). Considering the LOX activity as a representative example, the activity of the 1-MCP + LMF group was always similar to the activity of the initial sample, although it increased to 2584.9 U g^−1^ after 28 days of storage, which was similar to the activity of the control group after 21 days, but obviously lower than the activity of the control group after 28 days (Figure 4F).

### 3.5. Changes in PG, β-Glucosidase, and Cx Activity

As shown in Figure 5A, the PG activity of the peach fruits in the control group declined to a minimum value of 358.79 U g^−1^ after 7 days; however, it increased to 550.21 U g^−1^ by the end of the storage period. The β-glucosidase and Cx activities of the control group continuously increased and peaked at 711.25 and 836.11 U g^−1^, respectively, by the end of storage. However, the PG, β-glucosidase, and Cx activities were significantly lower in the 1-MCP + LMF group than in the control throughout the storage period (Figure 5B,C).

### 3.6. Changes in Fungal and Bacterial Community Structures

Fungal genera with relative abundances of > 1% were considered dominant fungi. As shown in Figure 6A and Appendix A, there were six original dominant genera in the fresh peach fruits: *Alternaria* (57.71%), *Stilbella* (21.35%), *Trichothecium* (6.94%), *Acremonium* (5.18%), *Aureobasidium* (3.12%), and *Holtermanniella* (1.74%). After 35 days of storage, the dominant fungal genera of peach fruits in the control and 1-MCP + LMF groups changed in both abundance and number. The dominant genera in the control group increased to 12: *Alternaria* (31.74%), *Aureobasidium* (27.09%), *Issatchenkia* (10.39%), *Coprinus* (4.21%), *Coprinellus* (3.45%), *Cladosporium* (3.38%), *Sodiomyces* (2.22%), *Talaromyces* (2.22%), *Trichoderma* (1.94%), *Pyrenochaetopsis* (1.46%), *Murakia* (1.18%), and *Rhodotorula* (1.16%). However, there remained six genera in the 1-MCP + LMF group, although the composition changed to *Aureobasidium* (81.39%), *Alternaria* (9.47%), *Holtermanniella* (2%), *Meyerozyma* (1.53%), *Tetraspora* (1.4%), and *Golubevia* (1.07%).

As shown in Figure 6B and Appendix A, *Streptophyta* was the only dominant bacterial genus (88.85%) with a relative abundance greater than 1% at the beginning of storage. In the control group after 35 days of storage, *Streptophyta* remained the only dominant bacterial genus, but its relative abundance declined to 66.05%, indicating that the peach bacterial community structure was generally stable. The 1-MCP + LMF treatment impacted the peach bacterial community, such that both *Streptophyta* (49.83%) and *Pantoea* (15.12%) were the main dominant bacteria after 35 days of storage.

## 4. Discussion

The appearance and nutritional components of fruits are important considerations when evaluating their commercial value. After fruits are harvested, they remain alive and undergo vigorous physiological and biochemical metabolic processes during storage; these processes affect their nutritional components, as well as their color, aroma, taste, and texture. If changes relating to sensory quality parameters exceed specific thresholds, fruit senescence or rot will occur, and the fruits will lose their commercial value [20]. In addition, most fruits lose weight during storage; excessive weight loss causes obvious epidermal shrinkage, wilting, and reduction of tissue hardness [21]. Appropriate postharvest treatments effectively decrease the physiological metabolism, reduce water loss, and maintain a generally stable nutrient content and appearance [22]. Kartal et al. [23] showed that a polypropylene microporous membrane (pore size 90 μm, hole number 9) combined with a deoxidizer effectively maintained the pH, SS content, electrical conductivity, and sensory quality of strawberries at 4 °C; it extended the shelf-life of strawberries longer than 4 weeks. Microporous membranes have been certified to be available to extend the shelf-life of grapes [10]. In the present study, “Feicheng” peach fruits in the 1-MCP + LMF treatment group lost only 4.67% of their weight after 28 days of storage; their nutritional components and appearance were maintained. Moreover, 1-MCP + LMF effectively inhibited the increases in SS and RS contents, while decreasing the TA and AsA contents of “Feicheng” peach fruits during storage. These results suggested that 1-MCP + LMF had an obvious preservation effect on peaches. Compared with the control treatment, 1-MCP + LMF significantly reduced the respiration rate, weight loss, and rot rate of peach fruits. Moreover, the combined treatment suppressed the increases in the SS and RS contents and the decreases in the TA and AsA contents.

Fruit softening affects the storage performance and shelf-life of peaches after harvest [24]. Softening usually accelerates the deterioration of peaches. Fruit softening is mainly caused by the degradation of cell wall materials, such as cellulose, hemicellulose, and protopectin [25]. Polygalacturonase, β-glucosidase, and Cx are the key enzymes that decompose pectin and cellulose. Inhibition of PG, β-glucosidase, and Cx enzyme activity can delay the decrease in fruit firmness and postpone softening [26,27]. Zhang et al. [27] showed that 1-MCP effectively inhibited the declines of PG, β-galactosidase, Cx, and pectin methyl esterase activities in nectarines during storage; it maintained high firmness. In our present study, 1-MCP + LMF treatment significantly inhibited PG, β-glucosidase, and Cx activities, meanwhile the co-application preserved hardness and maintained a high protopectin content and low soluble pectin content; these findings were generally consistent with the sensory evaluation results.

In addition, the oxidative damage of cell membranes caused by reactive oxygen species (ROS) affects fruit hardness and causes fruit softening [28]. ROS can be generated from respiratory metabolism, then promote membrane lipid peroxidation by activating membrane lipid metabolism-related enzymes such as lipase and LOX [29]. To remove ROS, an enzymatic scavenging system has evolved in plants. Various enzymes, including superoxide dismutase, CAT, and APX, are important in the scavenging system. Improvements to superoxide dismutase, CAT, and APX activities are reportedly beneficial for lightening cell membrane damage by maintaining the dynamic balance of ROS production and removal [30]. Song et al. [29] found that 1-MCP fumigation maintained high superoxide dismutase and APX activities; attenuated the chloroplast POD, lipase, and LOX activities; maintained membrane integrity; and ultimately alleviated cabbage yellowing. In our study, 1-MCP + LMF treatment maintained the ROS scavenging ability, with high CAT and APX activities; it also restrained membrane lipid peroxidation, with low lipase and LOX activities in peach fruits during cold storage. These changes reduced O_2_^−^ production and helped to scavenge H_2_O_2_. Our results suggested that 1-MCP + LMF could balance ROS metabolism and restrain membrane lipid peroxidation during the cold storage of peach fruits.

Infection by microorganisms is a major reason for the rapid decay of peach fruits after harvest; fungal infection is the main concern. The major postharvest pathogens of peaches include *Botrytis cinerea*, *Penicillium expansum*, *Rhizopus stolonifer*, and *Monilinia* spp. [31]. Zhang et al. [20] identified *Alternaria*, *Botryosphaeria*, *Penicillium*, and *Fusarium* in the fruits of the “Doyenne du Comice” pear. Zhang et al. [32] isolated six fungi from the peach fruits, including *Alternaria tenuis*, *B. cinerea*, *Penicillium digitatum*, *Rhizopus nigricans*, *Trichothecium roseum*, and *Aspergillus niger*; *A. tenuis* and *B. cinerea* were the main pathogenic fungi responsible for fruit rot. In our present study, high-throughput sequencing technology was performed to assess the abundance of the main fungi and bacteria in “Feicheng” peaches. There were six dominant genera in the initial samples at the beginning of storage: *Alternaria*, *Stilbella*, *Trichothecium*, *Acremonium*, *Aureobasidium*, and *Holtermanniella*. The sum of the relative abundances of these genera was 96.04%. The dominant genera identified in this study were consistent with the dominant genera identified in other studies, although there were slight differences in composition among production regions. *Alternaria* was detected in our research and has also been observed in peach fruits from other production regions [32]. It is considered a typical spoilage fungus that must be prevented and controlled during storage. The iron content in fruit is reportedly an important factor in *Alternaria* virulence [33], but *Aureobasidium* strains L1 and L8 compete for iron with *Monilinia laxa* through the secretion of siderophores. Thus, *Aureobasidium* is considered a potential biological control to reduce peach fruit rot [34]. Our results confirmed that 1-MCP + LMF increased the relative abundance of *Aureobasidium* to 81.39%. Besides, 1-MCP + LMF increased the relative abundance of *Holtermanniella,* which is considered a potential biological control agent to prevent *Fusarium scab* in wheat by competing with *Fusarium* [35]. However, its antagonistic effect on other fungi in peach fruits requires further confirmation.

Some postharvest treatments (e.g., ozone, hot water treatment, and 1-MCP treatment) substantially delay fruit deterioration by inhibiting fungal exuberate [27,36,37]. Zhang et al. [32] found that the strain *Bacillus subtilis* JK-14 showed significant antagonistic activity against the growth of *A. tenuis* and *B. cinerea*; it effectively controlled fungal diseases in peach fruit. Furthermore, 1-MCP reduces the cumulative incidence and severity of *Alternaria alternata*, *B. cinerea*, and *Fusarium* spp. in tomato fruit [38]. Microporous film packaging inhibits fungal rot in cherry fruits by maintaining a high CO_2_ pressure (10–20 kPa) [39]. In the present study, 1-MCP + LMF treatment significantly reduced the relative abundances of pathogenic fungi (e.g., *Alternaria*, *Stilbella*, *Trichothecium*, and *Acremonium*) in peach fruits after 35 days at 5 °C, while it increased the relative abundances of antagonistic fungi (e.g., *Aureobasidium* and *Holtermanniella*). The combined treatment also significantly reduced the rot rate to 6.25%, indicating that 1-MCP + LMF treatment alleviates decay in peach fruits by inhibiting fungal growth and propagation. The observed preservation effect could be attributed to synergistic effects from the co-application of 1-MCP and LMF; 1-MCP reduces respiration and physiological metabolism in peach fruits, while a permeable LMF provides a favorable gaseous environment, with high CO_2_ and low O_2_ in the packaging.

## 5. Conclusions

The co-application of 1-MCP and LMF significantly inhibits the respiration, reduces the degradation of cell wall materials, and delays ROS accumulation and membrane lipid peroxidation in honey peach fruits during cold storage; these changes can slow the softening of peach fruit. The combined treatment also improved the microbial community structure, while reducing the weight loss and incidence of rot, thus maintaining good sensory and nutritional qualities in peach fruit.

## Figures and Tables

**Figure 1 foods-11-01733-f001:**
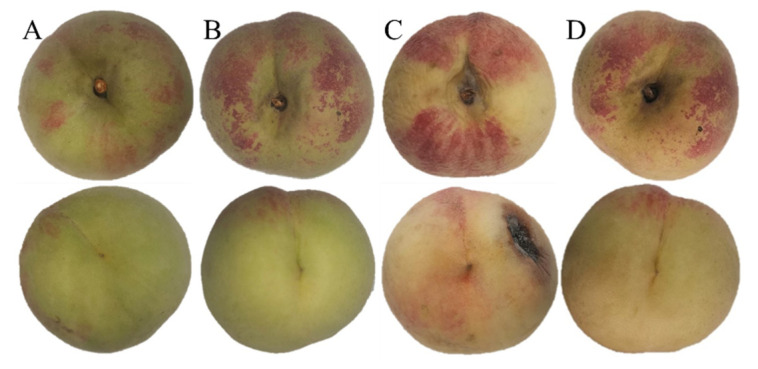
Appearance of peaches before and after storage at 5 °C for 35 days. (**A**), control at time 0; (**B**), combination at time 0; (**C**), control after 35 days; (**D**), combination after 35 days.

**Figure 2 foods-11-01733-f002:**
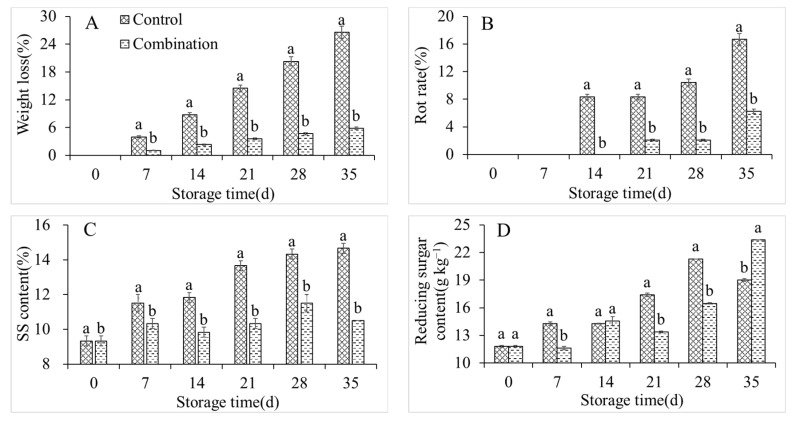
Weight lost (**A**), rot rate (**B**), SSC (**C**), reducing sugar (**D**), TA (**E**), AsA (**F**), propectin (**G**), and soluble pectin (**H**) contents of peaches during storage at 5 °C. Vertical bars indicate the standard errors of three replications. Different lowercase letters at same storage time points are significantly different (*p* < 0.05).

**Figure 3 foods-11-01733-f003:**
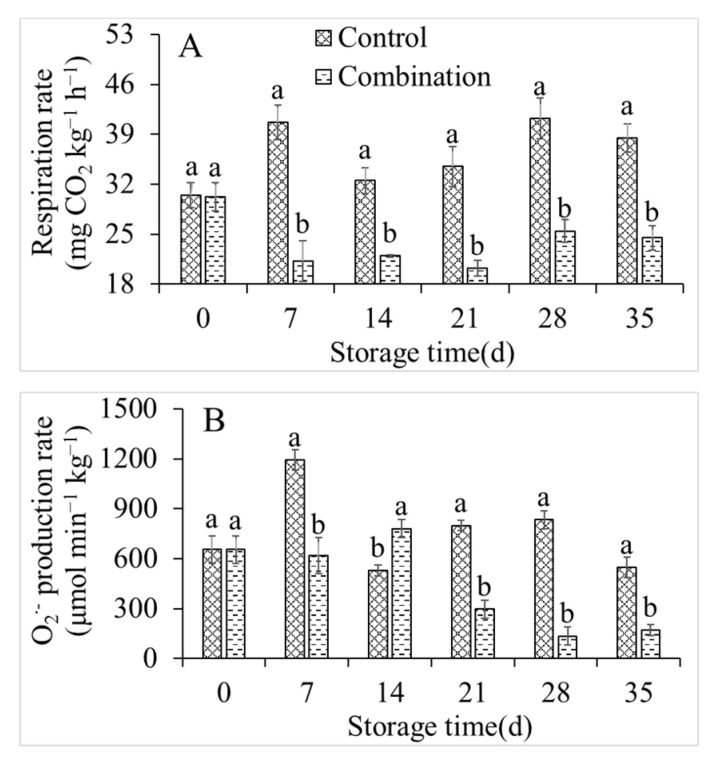
Respiration rate (**A**), O_2_^−^ production rate (**B**), and H_2_O_2_ content (**C**) of peaches during storage at 5 °C. Vertical bars indicate the standard errors of three replications. Different lowercase letters at same storage time points are significantly different (*p* < 0.05).

**Figure 4 foods-11-01733-f004:**
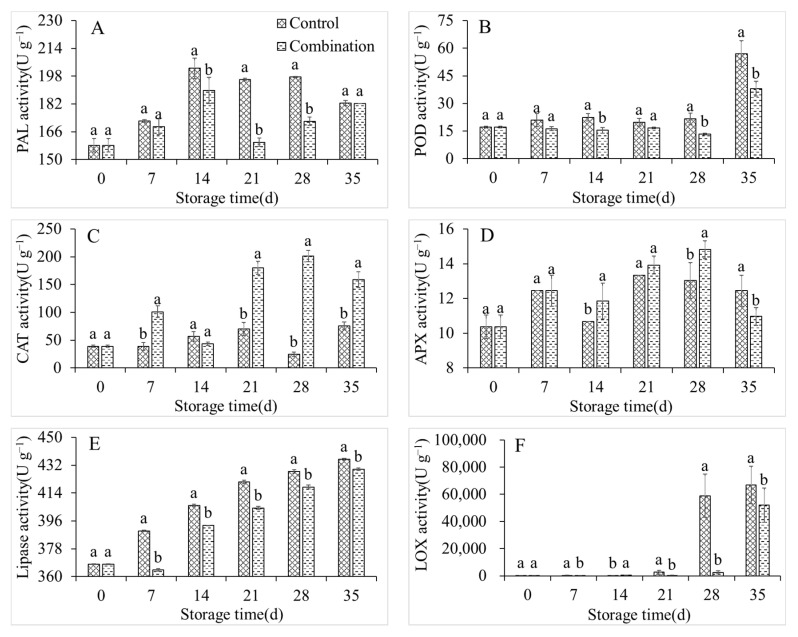
PAL (**A**), POD (**B**), CAT (**C**), APX (**D**), lipase (**E**), and LOX (**F**) activities of peaches during storage at 5 °C. Vertical bars indicate the standard errors of three replications. Different lowercase letters at same storage time points are significantly different (*p* < 0.05).

**Figure 5 foods-11-01733-f005:**
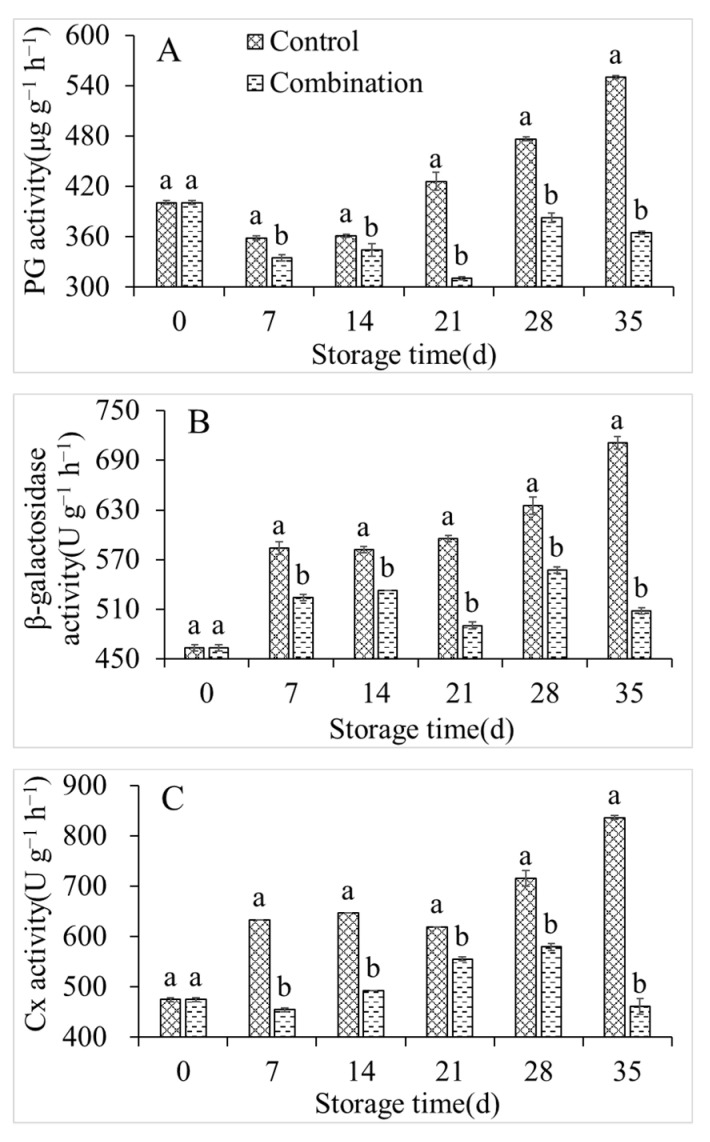
PG (**A**), β-galactosidase (**B**), and Cx (**C**) activities of peaches during storage at 5 °C. Vertical bars indicate the standard errors of three replications. Different lowercase letters at same storage time points are significantly different (*p* < 0.05).

**Figure 6 foods-11-01733-f006:**
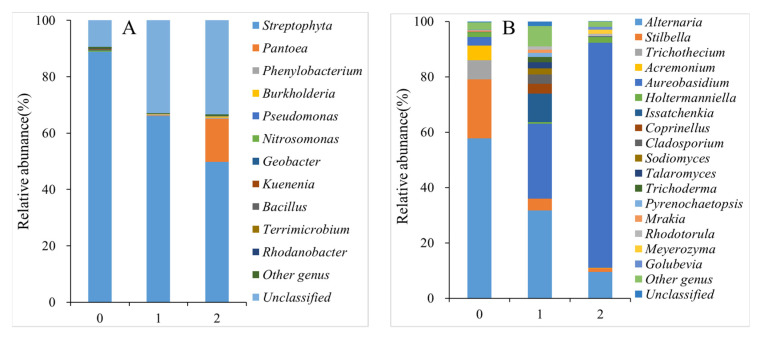
Bacterial (**A**) and fungus (**B**) diversity of peaches. 0, control at time 0; 1, control after 35 days; 2, combination after 35 days.

## Data Availability

All data supporting this research result are displayed in the paper and/or the Appendix A. The raw data of the transcriptome and metabolome analysis may be requested from the authors.

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
