# Peer review of "Co-Application of 1-MCP and Laser Microporous Plastic Bag Packaging Maintains Postharvest Quality and Extends the Shelf-Life of Honey Peach Fruit"

_foods, 2022, doi:10.3390/foods11121733_

Round 1

Reviewer 1 Report

I had the opportunity to read and review the manuscript entitled “Co-application of 1-MCP and laser microporous plastic bag packaging maintains postharvest quality and extends the shelf-3 life of honey peach fruit”.

A major objective of the manuscript is to investigate the synergistic preservation effect of 1-methylcyclopropene (1-MCP) and laser microporous film 18 (LMF) on shelf life of peach fruit. I find the manuscript to be interesting and within the scope of the journal. The relevant aspects of the topic are present. This study contributes to the development of a novel methodology for maintains postharvest quality and extends the shelf-life of honey peach fruit, as well as providing biochemical parameters for monitoring and assessing the quality of peach fruits during storage.

The experiments were well designed and described in detail, the research methodologies were scientifically sound and the data were appropriately analyzed and interpreted. the results were clearly presented in graphs and text. The English grammar and language were also good.  I have very few editorial suggestions for your consideration. I have indicated my editorial suggestions on attached PDF file.

Author Response

Thanks a lot for your comments and kind suggestions of our manuscript (No: foods-1738159). We provide this cover letter to explain, point by point, the details of our revisions in the manuscript and our responses to your comments as attachment.

Reviewer 2 Report

In this manuscript, the authors tested the effects of 1-methylcyclopropene (1-MCP) in combination with laser microporous plastic films (LMF) on fruit quality and fruit metabolism during storage for 35 days at 5°C compared with control. The authors reported synergetic effects for 1-MCP and LMF on maintaining fruit quality and reducing total fruit loss.

The manuscript idea is good but unfortunately  it lakes the correct experiment design by ignoring testing the individual treatments of 1-MCP or LMF compared with control or the combination to give correct scientific results if there is a synergetic effect of the treatments or not. Without testing individual treatments, the results and recommendations could not be adopted because the effect might be related to only 1-MCP or LMF and by the applying of two treatments, the cost will be high without scientific evidence.

I suggest running the experiment for one more year including individual treatments and present the data from two years to confirm the recommendations.

Author Response

(The authors gave the same response as above.)

Reviewer 3 Report

The manuscript is well written, but a few points need elucidation.

1. Introduction, last paragraph, please state the reason of conducting this research, what is the missing information based in the literature that preceded. Also, state the design to obtain this information. The results in this paragraph should be omitted.

2. In M&M, plant material, please describe the harvest maturity indices by which harvest was conducted (degree of ripeness).

3. Determination of weight loss, rot rate, and respiration rate, please describe the method of measurement. Set up, no of fruit etc.

4. Statistics, please make clear the method by defining the replications used (3 reps each consisting of pooled 5 fruit, or 15 fruit replications?)

5. In fig legends please indicate what the letters a &b stand for.

6. By viewing the protopectin figure  one realizes that here is a great variability among samples.  The attempted interpretation does not give any insight. Personally, I think that there is great variability among fruit in regard to harvest maturity and that his variability is masked by pooling 5 samples in one replication. Is this the case for rot incidence? I assume that all measurements suffer by this factor in addition to sampling by taking but a few grams of peach flesh. Based on these observations I would appreciate a relevant discussion of the manuscript.

Author Response

(The authors gave the same response as above.)

Round 2

Reviewer 2 Report

Based on my first revision for the manuscript my decision was reject the manuscript because in my point of view the data need more work for other year to assess the effects of all individual treatments on fruit quality and shelf life. For that, I do not have anything to add.